# The Exchange of Gifts between Christians and Muslims on Lusignan and Venetian Cyprus 1192–1517

Nicholas Coureas 

Department of History, Cyprus Research Centre, Nicosia 1095, Cyprus; ncoureas@hotmail.com

**Abstract:** In this paper which comes under the theme of macro-historical perspectives and Mediterranean history I shall discuss the exchange of gifts in three sections: first between the Lusignan kings of Cyprus and the sultans of Mamluk Egypt and Syria, between the Lusignan kings and the Turkish emirs of Anatolia, and thirdly between the Venetian rulers of Cyprus, including Queen Catherine Cornaro, and the Mamluk sultans for the period postdating 1473. Many of these exchanges of gifts took place during times of war, sometimes during the prelude to hostilities and sometimes immediately after their end. In addition, exchanges of gifts between Christian and Muslim rulers occasionally took place at times of peace. The reasons why gifts were exchanged, the type of gifts exchanged when these are described, not an invariable occurrence, and the symbolism underlying these exchanges of gifts will also be analyzed where possible. Furthermore, occasions when the recipient refused to accept the gift and why it was rejected shall also be examined. Where possible, comparisons with the exchange of gifts taking place in other societies and countries of the Eastern Mediterranean will be alluded to and discussed. In terms of source materials, the Cypriot chronicles of the fifteenth and early sixteenth centuries will form the principal source but reference shall also be made to diplomatic correspondence of the period under discussion. The exchanges of gifts normally took place within a diplomatic setting, this being the arrival of embassies, and gifts were sent sometimes to the close advisers of a ruler who could influence his policy and decision-making. On certain occasions, however, the exchanges of gifts could take place outside of this diplomatic context, for example as an expression of gratitude for assistance received.

**Keywords:** Mamluks; Turks; Anatolia; Alexandria; Venetians; gifts; robes; saddles

## 1. Introduction

To date there has been no comprehensive study on the history of the exchanges of gifts, conducted usually but not invariably within the context of international diplomacy, between the Lusignan kings of Cyprus, the Venetian rulers who succeeded them and their Muslim counterparts in Anatolia, Syria, and Egypt. The importance of gift-giving in international relations has been addressed by several scholars, such as Doris Behrens-Abouseif in her monograph on practicing diplomacy in the Mamluk Sultanate, with its emphasis on gifts and material culture in medieval Islam, as well as by Gerasimos Merianos in his article on the role of gifts in Byzantine diplomatic relations with the various powers to the East and West of the Byzantine state (Behrens-Abouseif 2016; Merianos 2007, pp. 199–222). In his article on embassies and ambassadors in Mamluk Cairo Yehoshua Frenkel observes that 'gifts are powerful symbols in public life and instruments that fortify social and political networks ... instrumental in facilitating communications between individuals and parties, strengthening mutual ties and cementing friendship.' Ambassadors to Cairo, the Mamluk capital, were routinely handed gifts at welcoming receptions and generally brought gifts with them for the Mamluk sultans. These were an impressive variety of presents encompassing slaves, textiles, silks, gems, animals, tents, keys to conquered fortresses and on one occasion a clock with figures playing musical instruments (Frenkel 2019, pp. 251–55). The gifts exchanged between the Christian rulers of Cyprus, Franks,

and Venetians, with their Muslim counterparts are less varied and sometimes unspecified. The source material for recording these exchanges of gifts is patchy, covering the later fourteenth and fifteenth centuries, and the early sixteenth century but not the thirteenth and the first half of the fourteenth century, the high point of Lusignan rule. Nonetheless, since the gifts were generally given or received with specific political objectives in mind, the practice of exchanging them is worth discussing.

As the preceding paragraph makes clear, behind the act of gift-giving lay the principle of reciprocity, an idea going back in time to archaic societies. In his monograph on exchange in archaic societies, Marcel Mauss has made clear that in such societies the very concept of a free gift was wrong. This was because 'Refusing requital puts the act of giving outside any mutual ties . . . that is what is wrong with the free gift. A gift that does nothing to enhance solidarity is a contradiction.' He has also observed that 'it is not individuals but collectivities that impose obligations of exchange and contract upon each other' (Maus 2002, pp. ix–x and 6). Maus had in mind the exchange of gifts within specific societies, not their exchange between states. Nevertheless, the concept of reciprocity will become clear in this article. The Lusignan kings of Cyprus offered presents to the Mamluks to secure peace, the Turkish emirates of Southern Anatolia offered King Peter I of Cyprus gifts to avert war and the Mamluk sultans of Egypt and Syria offered gifts, especially robes, to secure the vassalage of the recipient. Therefore, regarding reciprocity, there is a strong continuity between the archaic societies and the eastern Mediterranean in the Later Middle Ages.

## 2. Gifts Exchanged between the Kings of Cyprus and the Mamluk Sultans

The fifteenth-century Cypriot chronicler Leontios Makhairas narrates how following his purchase of Cyprus in 1192 from King Richard I of England, who had conquered it in 1191 from the rebel Isaac Comnenus, Guy de Lusignan, founder of the Lusignan dynasty, allegedly solicited the advice of Saladin on how to retain Cyprus. Saladin replied: 'My son, give all to gain all . . . give abundant gifts and bring to yourself great men' (Dawkins 1932, vol. 1, pp. 22–23, sec. 25). It is highly unlikely that Saladin, who had reconquered Jerusalem and considerable territory from the Franks in 1187, depriving Guy of his kingdom, was ever asked for advice. The story shows, however, that Makhairas considered gift-giving by the island's Lusignan rulers to have begun at the very outset of the dynasty's rule, on the advice of a Muslim ruler who was a precursor of the Mamluk sultans, who ruled over Egypt, Syria, and the Holy Land from 1250 until 1517. Nevertheless, the examples of gift-giving Leontios Makhairas gives in his chronicle all date from the second half of the fourteenth and the first half of the fifteenth centuries.

The first recorded incidence of gift-giving took place in 1363 when the Turks of southern Anatolia, exploiting the absence of King Peter I in Europe and the losses Cyprus had suffered on account of visitations of the plague, began raiding the island by sea. Sir John de Sur the admiral of Cyprus retaliated by raiding southern Turkey. When one of the Turkish raiders, Reis Muhammad Pasha, learned that the Lusignan galleys were searching for him, he hid in the Syrian port of Tripoli, subject to the Mamluk sultanate. The admiral then asked Melek the emir of Tripoli not to protect Reis Muhammad because he was an enemy of Cyprus, whereas Cyprus and the Mamluk sultanate were at peace. Melek, who seems to have sympathized with Reis Muhammad, declared that he could not expel him without the sultan's permission, so Sir John de Sur sent a galley with Melek giving him two Saracens to take him to the Mamluk Sultan (Dawkins 1932, vol. 1, pp. 124–27, sec. 144). Before the galley departed for Egypt, the admiral had presents sent to Melek, but the latter on seeing Turkish captives in chains on board the galley was angry. Not allowing the galley to leave for Egypt, he compelled John de Sur to return to Cyprus. Clearly, the admiral had given the emir Melek presents to gain his co-operation in obtaining the sultan's permission to expel Reis Muhammad from Mamluk Tripoli, but without success.

Following King Peter I's attack on Alexandria in October 1365 an embassy from Cyprus sent to make peace is recorded by Ibn Ayas and Al-Maqrizi. It apparently reached Cairo

with lavish gifts in March 1366. The Cypriot envoys, fearing reprisals from the population of recently sacked Alexandria, asked Sultan Shaban to provide hostages, to be held on board their ship until they had returned to it safely from their mission. Agreeing to this but worried that the Cypriots might execute the hostages sent, they sent convicts dressed up as rich Alexandrian merchants. To make the pretense more convincing, they also had women and children, supposedly the merchants' relatives, stand at the harbor bewailing the merchants' departure, fearful that they might not come back. The Cypriot envoys on reaching Cairo were received not by the sultan, who being an adolescent might not have impressed them, but by Yalbugha, the commander-in-chief of the Mamluk armies, acting on the sultan's behalf (Behrens-Abouseif 2016, pp. 102–4).

The envoys departed without achieving their aim to conclude a peace, but Yalbugha accepted their presents, keeping for himself a crystal ewer mounted in gold and a box with unknown contents, distributing the other presents among his entourage. This incident shows how gifts could be accepted even when negotiations failed, as well as providing us a rare, and therefore, valuable insight into the kinds of gifts given, for frequently the chroniclers simply refer to presents without stating what they were. It is noteworthy that crystal in general was a gift highly prized by the Mamluks. Gifts of crystal reached them from Venice, and even in 1310 when Mamluk glass production peaked, crystal was among the diplomatic gifts sent to the Mamluk sultans. One example is the set made up of a basin with a ewer made of *billawr*, the Mamluk term for rock crystal, mounted in gold and studded with precious stones. This was dispatched by the king of Armenia to the Mamluk sultan Al-Nasir Muhammad to congratulate him on his restoration to the sultanate in 1310 to begin his third period of rule (1310–1341) (Behrens-Abouseif 2016, pp. 104 and 111).

At this stage, the Venetians, alarmed at the damage caused to their trade, intervened, and persuaded the Mamluk sultan to conclude a peace with the king. Apparently pleased with their intercession, the sultan sent envoys to Cyprus in the company of the Venetians. Landing at Famagusta in May 1366 and bringing great presents with them the Mamluk envoys journeyed to Nicosia, arriving in June and giving King Peter the presents along with letters from Sultan al-Ashraf Shaban. By way of reply, the king decided to send his own envoys to Cairo, with a letter of reply to the sultan's letter, 'fitting presents to give to the sultan, and many other presents' (Dawkins 1932, vol. 1, pp. 160–65, sec. 181–85). Ultimately, the negotiations for peace between both parties failed, but here as previously, one sees that gifts were exchanged with the aim of creating goodwill and facilitating negotiations. The reasons why peace was not achieved are discussed immediately below.

Intermittent warfare punctuated by negotiations continued. Following a Cypriot raid on Tripoli in January 1367, Sultan Shaban released a hitherto imprisoned Cypriot envoy, James de Belonia, dispatching him to Cyprus in February. He was accompanied by the sultan's own envoy who bore suitable presents and letters to King Peter I, asking him to conclude peace. The king at the time, mindful of the costs of the war, wished with the nobles in his council to make peace. This was not achieved. According to Leontios Makhairas the war party in Cairo was in the ascendant at the time. In fact, Emir Asandamur, who had seized power early in 1367, resumed negotiations with Cyprus, but King Peter seems to have thought that they wished to postpone the conclusion of peace to continue the war (Dawkins 1932, vol. 1, pp. 172–77, sec. 192–93 and 196; Onimus 2022, pp. 264–65). In May 1367 Sir James de Nores the turcopolier of Cyprus finally visited Cairo to conclude a peace. To smooth over the obstacles that had arisen, Sir James summoned a Genoese convert to Islam who happened to be a friend of his. He gave him suitable presents for the convert to grant both to the sultan and to the emirs who could be useful in bringing about peace. The convert persuaded the emir who had been urging the sultan to reject peace to now urge the opposite, probably by giving him presents, although this is not stated explicitly. Eventually, the sultan dismissed the Cypriot envoys 'with good grace' according to Leontios Makhairas, although peace was still not concluded, an indication that factions in the sultanate opposed to peace continued to be powerful (Dawkins 1932, vol. 1, pp. 182–87, sec. 202–3).

As late as August 1368 peace was still not concluded, largely because of conflicting factions within Cairo, some desiring peace, and others war. The account of the chronicler Leontios Makhairas makes this clear when recounting the treatment meted out in Cairo to the Genoese envoy Cassan Cigala, who finally returned to Cyprus in the autumn of 1368 to report to King Peter I that his attempts to conclude peace had failed. The king, clearly attempting to mend relations with the sultan, wrote him a letter and sent him two Saracen slaves, probably captives, who had been found in Cyprus (Dawkins 1932, vol. 1, pp. 206–11, sec. 225–28 and 230). Despite this overture, peace between Cyprus and the Mamluk sultanate was concluded only at the end of September 1370, nearly two years after King Peter's assassination by his own nobles in January 1369. Following the conclusion of this peace, Prince John of Antioch, the brother of the late King Peter and regent of Cyprus on behalf of King Peter II (1369–1382), who was still a minor, had the peace proclaimed throughout Cyprus. He received the letters and presents sent by the sultan and gave in exchange other letters and gifts to be taken to the sultan. In addition, he gave the sultan's envoys suitable presents and dismissed them, following which they returned to Alexandria (Dawkins 1932, vol. 1, pp. 290–97, sec. 305–9). Here one sees gifts between the Cypriots and the Mamluks being exchanged after the conclusion of peace, although the types of gifts are unfortunately not stated. In this instance, the presents were exchanged not to promote the conclusion of a peace, but to celebrate a peace already accomplished.

The next recorded exchange of gifts between Cypriots and Mamluks took place in the early fifteenth century. Following a period of raids and pillaging, with both the Cypriots and Mamluks conducting raids by sea on each other's territories, seizing captives, burning lands, and pillaging castles, peace between the two parties was concluded under King Janus of Cyprus (1398–1432) and the Mamluk sultan al-Muayyad Shaykh (1412–1421) (Ouerfelli 2004, pp. 330 and 332–34). King Janus sent his envoy Sir Thomas Provosto to the Mamluk sultanate in 1414, with the sultan receiving him honorably and granting him rich presents. After that, the sultan sent his dawadar to Cyprus as his envoy, accompanied by Sir Thomas. King Janus received him in turn honorably and peace was concluded, although it did not last long (Dawkins 1932, vol. 1, pp. 628–29, sec. 646). Cypriot raids on the Mamluk coastlines carried out to take captives who were either sold or forced to work in the Cypriot sugar plantations, continued. On the orders of Sultan Barsbay (1422–1438) Mamluk fleets carried out retaliatory raids against Cyprus in September 1424 and August 1425, but following these raids a Saracen emir named Muhammad ibn Khudaidar, called Seth in the anonymous chronicle of 'Amadi,' decided to intervene to stop further hostilities (Dawkins 1932, vol. 1, pp. 630–37, sec. 651–59; Hill 1940–1952, vol. 2, p. 474; Coureas and Edbury 2015, p. 456, sec. 1054; Darrag 1961, pp. 243–49).

This emir heard about King Janus through the ambassadors that the king had sent to Cairo. These ambassadors, Sir John Podocataro and the merchant Sir Thomas Prevost had befriended Sheikh Muhammad and along with others told him of the power, goodness and virtue of King Janus, so that the sheikh esteemed him, regarding him like a son. These ambassadors also offered Sheikh Muhammad many presents on the king's behalf, with the sheikh, who was very wealthy and of good character, accepting none of them except for some things to eat. To help the king and prevent a Mamluk invasion of Cyprus, Sheikh Muhammad sent his son to Cyprus with letters to seek an audience with King Janus and to induce him to cease the raids against Mamluk territories. Otherwise, the sultan, who was far more powerful than he, would attack and ruin Cyprus. The sheikh stressed that he could now do this more easily, because he had subjugated all the rebellious emirs of his domains, now united under a single rule. This is an important point, for under King Peter I the sultan had been a minor and disunity prevailed between those emirs desiring war and the others wanting peace (Dawkins 1932, vol. 1, pp. 638–41, sec. 661–62; Coureas and Edbury 2015, pp. 456–57, sec. 1054–55).

Sheikh Muhammad's son reached Famagusta carrying his father's letters addressed to the king as well as valuable presents. Nevertheless, he was prevented from seeing the king by the latter's councillors who sent two intermediaries, the king's squire Peter Pelestrin

and his physician John Synglitikos to speak to the sheikh's son at Lefkoniko, a locality near Famagusta. The king also sent men to serve the sheikh's son 'with white bread, and with white wine such as the king drank, and with cordials, and with many other excellent provisions.' The sheikh's son, however, not fobbed off by these presents, gave Pelestrin 100 ducats for the latter to arrange an audience with the king. Pelestrin journeyed to Nicosia informing the king of this, but could not arrange an audience because the knights in the king's entourage were opposed to it. The sheikh's son then gave Pelestrin Sheikh Muhammad's letters addressed to the king, instructing him to go back to Nicosia and give them to him, and to tell the king what the sheikh wished to convey to him. Pelestrin duly went to Nicosia, appearing before the king and his council. The councilors would not allow the king to obtain the letters, which they had translated into Greek and read out. They persuaded the king to reject Sheikh Muhammad's peace initiative because it revealed, according to their advice, the sultan's fear of the king's power. Agreeing with them, the king sent presents to the sheikh's son and an answer in the negative, after which the sheikh's son returned to Mamluk territory (Dawkins 1932, vol. 1, pp. 640–47, sec. 663–66; Coureas and Edbury 2015, p. 457, sec. 1056).

This episode highlights the inefficacy of mutual gift-giving on this occasion. The presents the sheikh's son brought failed to secure him an audience with King Janus, the presents the king sent the sheikh's son at Lefkoniko failed to dissuade the latter from persisting in his wish to see the king and the presents the king gave the sheikh's son before his return to Mamluk territory failed to stop the Mamluk invasion of 1426. During the invasion, the king, captured after being defeated in battle, was dispatched to Cairo, and returned only upon payment of a huge ransom. He was also forced to pay the Mamluks an annual tribute and place his kingdom under Mamluk suzerainty, which lasted until 1517 when the Ottoman Turks conquered the Mamluk sultanate (Holt 1986, pp. 185–86). The vassalage of the kingdom of Cyprus to the Mamluk sultanate was itself stressed, symbolized by gifts. According to Salih ibn Yahya, who took part in the expedition of 1426, commanding an old brigantine that had been constructed at Beirut, when the sultan freed King Janus on payment of the ransom demanded he granted him a robe of honor lined with ermine, as well as a horse with a golden saddle and a golden bridle. The sultan granted him more gifts just before his departure for Cyprus via Alexandria. These gifts, especially the robe, were not accidental, for the receipt of robes in the Muslim world signified the recipient's vassalage towards the donor (Mansouri 2001, pp. 102–3 and 109–10).

This practice is apparent in a wider geographical context. In early 1289 Armenian ambassadors of King Hetum II, who had become king after the death of King Leo II, arrived at Tripoli, which the Mamluks were besieging, and were well received. They were given robes of honor and instructed to tell King Hetum II to surrender the cities of Marash and Behesni, to continue paying the annual tribute that the Armenians had agreed to pay the Mamluks in 1285 and to keep to other unnamed conditions. A more unusual exchange of gifts took place between King Leo IV of Armenia and the Mamluk sultan Al-Nasir Muhammad in 1329. Leo had become king by overthrowing the regent Oshin of Korykos, whom he assassinated along with the regent's brother, Kostandin II of Lampron. To underline his seizure of power, Leo sent Oshin's head to the Mamluk sultan, who by way of recognizing his rule instructed the emir Shihab al-Din Ahmad to grant King Leo IV in return the sultan's livery, a sword, and a horse with a saddle and a bridle (Stewart 2001, p. 72; Chevalier 2009, pp. 637–38).

The dispatch of annual tribute did not end the practice of mutual gift-giving between Mamluk and Cypriot rulers. In 1427, just one year after Cyprus had been placed under Mamluk suzerainty, King Janus sent seven of his dignitaries to the Mamluk lands to serve Sultan Barsbay as his soldiers, with two of them converting to Islam. Following King Janus's death in 1432 and the succession to the throne of Cyprus of his underage son King John II, Sultan Barsbay sent a delegation to install John as king and to demand arrears in the payment of tribute. Having sworn fealty to Sultan Barsbay, who was sent the arrears in tribute and a present, King John II was invested with a robe of honor, namely the sultan's

livery, which denoted the king's vassalage to him (Holt 1986, p. 186; Edbury 2013, p. 154; Behrens-Abouseif 2016, p. 104). Robes of honor could be rejected as well as accepted. In January 1436 Sultan Barsbay received an embassy from Shah Rukh, the son of the Mongol conqueror Timur Lenk who ruled over part of his father's dominions. Shah Rukh through this embassy offered investiture as the ruler of Egypt as well as robes to the sultan. Had the sultan accepted he would in effect have acknowledged himself to be Shah Rukh's vassal. Instead, he had the robes torn to shreds, having Shah Rukh's ambassador beaten and repeatedly ducked in a horse pond until he nearly died (Holt 1986, pp. 188–89).

On contacting him he gave him letters and greetings from King John II but also a gift
Following his ascent to the throne King John II continued to pay tribute regularly, but the expenses attendant on receiving Mamluk envoys to collect this tribute when they visited Cyprus annually, and giving them presents, were burdensome. Therefore, in 1436 the king sent an embassy to Cairo that included the Spanish nobleman Pero Tafur, who asked the sultan to cease sending Mamluks annually to Cyprus. This was because the king would continue making regular payments of tribute, but by sending Cypriot envoys to Egypt instead. Pero Tafur and the embassy succeeded in their mission, with Cypriot envoys bearing tribute being sent continuously to Mamluk Egypt down to the Ottoman conquest of 1517. Pero Tafur wrote an account of his travels, including his mission to the Mamluk sultan Barsbay. He was instructed, on reaching Cairo, to contact the sultan's chief interpreter, a former Jew from Seville named Saym who had converted to Islam (Nepaulsingh 1997, pp. 12–13; Darrag 1961, p. 266).

On contacting him he gave him letters and greetings from King John II but also a gift of 200 ducats. The king sent him this money because his father, the late King Janus, had told him to do so to express his gratitude for the assistance Syam had given him during his captivity in Cairo. Saym received Pero Tafur and his party cordially, lodging them in his own house. On the day Pero had an audience with Sultan Barsbay the latter acceded to his requests made on King John's behalf. Prior to his departure, he granted him a piece of clothing normally granted to the king of Cyprus as a mark of the sultan's lordship over him. This was a robe of rich oriental green and red fabric worked in gold and lined with patterns of ermine. On his way back to Cyprus Pero Tafur sojourned at the port of Damietta, where the local governor treated him honorably on account of a letter of recommendation the sultan's chief interpreter had furnished him with. In addition, the chief interpreter asked the local governor to find some cockatrice skin to send the king of Cyprus as a present, which was duly done. Clearly, on this occasion, the presents given by both sides facilitated and sealed a successful mission regarding the payment of tribute from Cyprus (Nepaulsingh 1997, pp. 15–17 and 19). In this respect, their exchange recalls the conclusion of peace between Mamluks and Cypriots in September 1370, discussed above.

On 29 November 1456 the Mamluk sultan Al Malik al Ashraf Abul Nasr Aynal (1453–1461), responded to the congratulations King John II had conveyed to him on his ascent to power and remitting arrears of tribute due from Cyprus. The letter alluded to the celebrations held in Nicosia to celebrate the sultan's accession and acknowledged that part of the tribute was already paid, in camlets. It also stated that at the king's own request, the sultan would write to the Ottoman Sultan Mehmet II, asking him to call off the raiders who were subjects of his and had been ravaging the kingdom of Cyprus with their attacks. The sultan also alluded to the gift of an expensive robe and of a fine horse from the sultan's stables for the king, together with a silver saddle for King John II, consigned to his ambassador, to whom he had also given sumptuous drapes and a horse. Clearly, these gifts, given to the kings of Cyprus and to their ambassadors on previous occasions, were also tokens of vassalage (Mas Latrie 1852–1861, vol. 3, pp. 73–75).

Following the death of King John II in June 1458 and the war of succession that took place between his legitimate daughter Charlotte and his illegitimate son James, both sides sent envoys to the Mamluk sultan, who at that time was Aynal. The Cypriot chronicler George Boustronios, a contemporary of the events he describes, recounts how James and his supporters reached Cairo at the end of 1458. In 1459, shortly after Queen Charlotte's marriage to Louis of Savoy, the royal council dispatched ambassadors to Sultan Aynal,

who went to Cairo taking many presents with them. They apparently gained the sultan's favor, but their death in Cairo on account of the plague raging there nullified their success. A second embassy headed by Peter Podocataro now made its way to Cairo, meeting the sultan and presenting gifts to him. On leaving the sultan's presence Peter Podocataro also approached the emirs close to him, bringing them presents and so persuading them to support Queen Charlotte and her husband Louis as sovereigns of Cyprus. George Boustronios observes that 'in line with custom' they ordered the robe to send it to the queen, the robe clearly being a symbol both of vassalage and of recognition of the vassal's right to rule a subject territory (Coureas 2005, p. 90, sec. 41; Mas Latrie 1886, pp. 392–93).

Peter Podocataro's efforts were ultimately unsuccessful. William Goneme, an Augustinian friar who was one of James's principal supporters himself visited the same emirs, taking with him Nassar Hous, a Circassian supporter of James who knew their language to act as interpreter. Negotiating with them throughout the night, he was able to outbid Charlotte's party, offering the emirs greater bribes and possibly an increase in the annual tribute paid by Cyprus. On William's instructions, James went in person to the venue where the emirs planned to bestow the robe of vassalage on the queen's ambassador. Just as they were about to place it on the ambassador's shoulders, the Mamluks suddenly exclaimed 'Long Live King James!' and seizing the robe placed it on his shoulders. Furthermore, they placed the queen's ambassadors and those of the Hospitaller Grand master under James's authority. They also gave him the robes previously granted to the first and the second of the queen's embassies and those granted to John Dolfin, the Hospitaller ambassador sent to Cairo by Grand Master James de Milly. Clearly, the official presents the queen's embassies had brought Sultan Aynal did not achieve their object because his own emirs had received greater bribes from James's own supporters in Cairo. The sultan offered presents to James, issuing orders for an invasion force to be readied and to accompany James to Cyprus, personally selecting the Mamluks who would take part in it (Coureas 2005, p. 91, sec. 42; Mas Latrie 1886, p. 393; Mansouri 2001, pp. 123–24).

Having secured the sultan's alliance, a Mamluk invasion fleet was prepared, bringing James, his supporters and a Mamluk force to Cyprus and disembarking them near Famagusta on 18 September 1460. Even at this stage, Queen Charlotte's husband Louis of Savoy tried to win the Mamluk admiral over. He sent his envoy Brother Christopher to the admiral with gifts consisting of oxen, slaughtered animals, chickens, loaves of bread, sugar 'and many other things' that the admiral simply gave to the foot soldiers, placing Brother Christopher into James's custody (Coureas 2005, p. 94, sec. 49; Mas Latrie 1886, pp. 397–98). With Mamluk assistance, James won the civil war against his half-sister Queen Charlotte by August 1464, and she spent the remainder of her life outside Cyprus. Nonetheless, James was forced in the autumn of 1464 to have Janibek, the leader of the Mamluk troops on Cyprus, and his forces surprised and massacred outside Famagusta, taken early in 1464. This was because they had been kidnapping good-looking Cypriot youths from their parents, sending them to Egypt to be converted to Islam and trained as Mamluks (Coureas 2005, pp. 112–13, sec. 88; Mas Latrie 1886, pp. 416–17; Edbury 2013, p. 193).

On learning of this massacre, Janibek's sister urged Sultan Khushqadam (1461–1467) to have James assassinated. But James, perhaps fearing such a development, sent an envoy to the sultan immediately, someone called 'Jacob the Frank' according to Taghri Birdi. This envoy, arriving in Cairo in March 1464 with great gifts, explained to the sultan that James had had Janibek and his Mamluks killed because the latter wished to seize Cyprus for himself. George Boustronios states that the sultan and his emirs liked James, disregarding Janibek's sister because James had many people in the Mamluk sultanate who were fond of him, to whom he also gave rich presents. Here one sees how the gifts James sent to the sultan and to his personal contacts in the Mamluk sultanate, who were probably highly-placed emirs at the sultan's court, staved off a potential conflict that the massacre of Janibek and his forces might have provoked (Coureas 2005, pp. 113–14, sec. 89; Mas Latrie 1886, p. 417; Edbury 2013, p. 194; Coureas 2019, p. 734). It is recorded that James sent another mission to Cairo, probably just before 1 September 1464, to announce the impending fall of

Kyrenia to the sultan. It was headed by his key supporter William Goneme, by now Latin archbishop of Nicosia, who arrived in Cairo with valuable presents, 1000 pieces of camlet and additional merchandise worth 20–25,000 ducats. William Goneme's embassy took place to secure Sultan Khushqadam's support for James at a time when Queen Charlotte was also making overtures to this sultan. It succeeded in its purpose, and Mamluk support for James continued until his death in 1473 (Mas Latrie 1852–1861, 3 vols., vol. 3, pp. 129–30 note 1; Hill 1940–1952, 4 vols., vol. 4, pp. 619–20).

### 3. Gifts Exchanged between the Kings of Cyprus and the Turkish Emirs of Anatolia

The relations between Lusignan Cyprus and the Turkish emirates of southern Anatolia were marked by intermittent warfare, usually in the form of raids and counter-raids, but also by commercial exchanges. Warfare peaked under King Peter I, who had an ambitious crusading agenda and engaged in protracted but costly warfare against both Anatolian Turks and Mamluks. In 1360 King Peter's forces took the Armenian port of Gorhigos at the request of its own inhabitants, who doubted the ability of the King of Armenia to protect them. Leontios Makhairas states that thenceforward King Peter sent forces and supplies to Gorhigos regularly to protect it from the Turks, adding that the port also enjoyed divine protection, 'the miraculous icon of the Virgin of Gorhigos.' According to the chronicler, this icon blinded the Grand Karaman, the Turkish emir of Karamania. Remaining sightless for many days, he eventually admitted that a lady from Gorhigos had blinded him. To placate the icon, he removed his army from the vicinity of the city and had fashioned three great candles of camphor-wax and three silver lamps that were hung in front of the icon. In addition, he gave it four measures of oil every year and many ducats. Once the lamps had been lit and chanting had taken place all night, cotton was rubbed on the icon the following day and then placed on his eyes, curing him forthwith (Dawkins 1932, vol. 1, pp. 100–3, sec. 114–15). This tale is unusual because the gifts here, given by a Muslim emir to a Christian icon, were rendered so that he could expiate himself from the punishment of blinding the icon had meted out on account of his aggression towards Gorhigos. In fact, as an instance of gift-giving it is unique in the context of Christian-Muslim relations in Cyprus during the Lusignan and Venetian periods.

In another incident taking place after July 1361, the Turkish emir of Adalia learned that King Peter I was equipping an expedition against him. He repeatedly sent emissaries to Cyprus to placate the king and persuade him not to send this expedition, but without success. When the emissaries, however, learned that the king was at a place called the Mills, they came there by ship to encounter him, giving him presents as well as letters. The king simply took them and set sail, reaching Adalia on 23 August 1361, taking the place after a siege (Dawkins 1932, vol. 1, pp. 106–7, sec. 120–21). Clearly, on this occasion, the presents failed to deflect the king from his purpose. On 8 September, moreover, the king set forth from Adalia with his army, coming before the coastal town of Alaya. Alarmed by this development, the emir of the town came out with some followers of him, declared his subjection 'and gave him a lordly present.' The Cypriot forces accepted the presents at the king's behest, and the king then departed, making his way to Monovgat. The emir of the place, unable to meet the king himself but anxious not to offend him, sent him his emissaries and a present. The king, returning the presents, conveyed his greetings to the emir, declaring that thenceforth he would consider him his own man. On these last two occasions, the presents sent to King Peter I by the emirs of Alaya and Monovgat appeased him, although they had to declare their subjection to him (Dawkins 1932, vol. 1, pp. 108–9, sec. 124–25). The presents, unlike those sent to or exchanged with the Mamluks, were not given to negotiate a peace, but to prevent hostilities, and so were of a pre-emptive and placatory nature.

A later passage in the chronicle of Leontios Makhairas indicates that the emirs of southern Anatolia gave gifts to King Peter I of Cyprus regularly to maintain peace with him. In the summer of 1367, the king departed from Rhodes to Adalia, sending his envoy Sir John Monstri to summon the emir Takka into his presence near Adalia. Takka came into

the king's presence. Having done him due honors and acknowledged his subjection he then departed, with the king likewise departing for Adalia. Meanwhile, various emirs of these regions sent their own envoys to the king with 'worthy presents . . .according to the custom' to confirm peace with him (Dawkins 1932, vol. 1, pp. 188–89, sec. 208). Clearly, the presents the king received from various Turkish emirs of southern Anatolia on a customary basis were in practice tokens of vassalage granted to maintain peaceful relations.

The emirs' vassalage came to an end in 1373 under King Peter II, who had succeeded his father Peter I as king in January 1369. In May 1373, in view of the imminent Genoese attack on Cyprus, it was decided that Adalia could no longer be held and that its garrison would be needed to defend Cyprus. Therefore, it was decided to deliver Adalia peacefully to the emir Takka, who in exchange would become the king's liegeman and swear to pay him tribute. The king's two envoys, Sir Baldwin Mistachiel a citizen of Famagusta and George Pissologos from Nicosia, duly met Takka and read out the king's letter. Takka accepted the proposals joyfully and gave the envoys great gifts, clearly given on this occasion by way of rejoicing. On 14 May 1373 Takka came and set up his camp in front of Adalia, taking the oaths that the king required from him. He also had presents sent to the king, which on this occasion are described in detail. They consisted of a set of silver vessels, and among these numerous vessels was a great silver cup of considerable weight and value, of the kind that the Turks used when celebrating. It functioned as a container for wine, which was then taken out with a ladle and given to the guests for them to drink. Shaped like a bowl with a foot, it apparently weighed eight marks (Dawkins 1932, vol. 1, pp. 344–45, sec. 366–68). With the evacuation of Adalia followed by the defeat of Cyprus in the war of 1373–1374 with Genoa, the kingdom could no longer keep the emirs of southern Anatolia in subjection.

The Lusignans were not the only Christian power exchanging gifts with the Turks of Anatolia during the fourteenth century. The Byzantines exchanged gifts with the Anatolian Turks, especially during the civil war that afflicted Byzantium between the years 1341 and 1347. At that time Turkish mercenaries were in great demand by both sides, each hoping through their employment to prevail against the other. One of the parties involved was John Kantakouzenos, who eventually prevailed in 1347, becoming emperor. The other party was Empress Anne of Savoy, the widow of the deceased Emperor Andronikos III, who had died in 1341, the Oecumenical patriarch of Constantinople John XIV Kalekas and Alexios Apokaukos, who supported Emperor John V, still a minor. When trying to detach Emir Umur Beg from his alliance with Kantakouzenos, Apokaukos offered him gifts and money. Likewise, when Emir Suleyman of Karasi met Kantakouzenos near Gallipoli to confirm their alliance, he offered him horses and weapons, receiving gifts in return. In the summer of 1346, the Saruhan Turks accepted gifts and money from Empress Anna, but nonetheless switched sides. Horses had a particular value as a present, being a symbol of prestige for both Byzantine and Turkish nobles and of practical value for the Turks. Kantakouzenos's wife Irene granted Umur and his troops 100 horses when they arrived at Didymoteichon in Thrace to assist Kantakouzenos, apologizing over not having been able to provide more. But whereas the Byzantines granted gifts to Anatolian Turkish emirs because they needed them, the Lusignans generally received gifts from the southern Turkish emirs in the context of vassalage, an important difference (Beihammer 2022, pp. 479–82).

## 4. Gifts Exchanged between the Venetians on Cyprus and the Mamluks

Following the death of King James II in July 1473, conspiracies by the supporters of Charlotte to restore her and by some of King James's former Catalan mercenaries to seize power and hand Cyprus over to King Ferrante of Naples were scotched. The Venetians by January 1474 were firmly in control of the island, a control underpinned by the garrisons they installed on Cyprus and by the presence of the powerful Venetian fleet. Fearful of growing Ottoman power and anxious to maintain good relations with the Mamluks, to whom Cyprus continued to be nominally subject and to pay an annual tribute, the Venetians sent envoys with tributes and gifts to them regularly. From 1474 to 1489 the Venetians

controlled Cyprus through Queen Catherine, a Venetian noblewoman of the Cornaro family and the widow of the late King James II, but following her abdication Venice imposed direct rule over the island.

Following the death of King James II his widow Catherine had to secure the Mamluk sultan Qaitbay's recognition of her rule over Cyprus. To this end, she sent as her emissary a burgess of Famagusta named Andrea Casoli to Cairo on 5 July 1473 to inform the sultan of the king's death. His mission was successful, with the sultan showing himself well disposed towards him and having him clothed in cloth of gold, which must refer to the robes of vassalage customarily bestowed by the Mamluks. He returned to Cyprus on 20 August 1473, with instructions from the sultan to have the tribute sent to Cairo. Another envoy sent from Cyprus to the sultan on 20 August 1473 was Andrea alias Anthony de l' Orsa. He too returned on 26 October having succeeded in his mission. He brought the message that the sultan, well disposed towards Cyprus, and by implication towards Queen Catherine as its ruler, should be sent 24,000 ducats in tribute for three outstanding installments, and a good present because the queen had come into possession of her kingdom (Coureas 2005, pp. 120, 122, 125 and 129, sec. 101, pp. 111–12, 120 and 149). The sultan clearly desired the present as a reward for bestowing his recognition regarding the legitimacy of Queen Catherine's rule.

Sultan Qaitbay gave presents as well as receiving them. On 5 May 1476, he sent Queen Catherine a letter acknowledging payment of two years' tribute from Cyprus that had been delayed on account of the troubles that she had overcome. This was an oblique reference to the failed Catalan attempt to overthrow her, as well as because of the damages the locusts had caused in Cyprus. Congratulating her and making known to her envoy, Thomas Ficard, his recognition of her as queen of Cyprus, he also released from prison an envoy she had sent previously, during the time of her troubles, permitting his return to Cyprus. He sent her various presents, a silk gown lined with ermine, four pieces of other silk textiles, a golden saddle, 14 pieces of Chinese porcelain, ten pounds in weight of aloe wood, fifteen pounds in weight of benzoic resin, ten boxes of theriac, and a flask of balsam oil. This gift differed from those the Mamluks sultans sent to their Muslim or even at times their Christian vassals because it lacked armor and weapons, perhaps because the gift of armor and weapons to a female ruler was deemed inappropriate. In this letter, the sultan stated that the ambassador had been clothed and had been paid his expenses. Regarding the present sent to her, the sultan expressed the wish that in accepting it she would wear the robe he had sent as a token of her obedience to him, to confound her enemies and to pray for his longevity. This letter is especially valuable, it lists in detail the presents the sultan sent but also his reason for sending them (Mas Latrie 1852–1861, vol. 3, pp. 405–6; Hill 1940–1952, vol. 3, p. 725 note 2).

There is one more recorded embassy to Sultan Qaitbay during Queen Catherine's reign, that of 1483. Thomas Ficard was the envoy, probably chosen on account of his success in 1476. Once again, the ostensible reason for sending him was over arrears in the payment of tribute. Ficard wrote a report of his embassy, dated 30 December 1483, to George Contarini the titular count of Jaffa. Recounting how the sultan detained him at first due to arrears in the payment of tribute, he continued that the queen then replied positively to the sultan's letters requesting its payment, while also requesting Ficard's release. This was accomplished by the dispatch of 'other presents and artifices,' an oblique reference to bribes granted to persons close to the sultan to facilitate Ficard's release. These persons, in turn, influenced the sultan, who on 25 September had Ficard and one of the translators accompanying him dressed in the customary robes of honor, denoting vassalage. The Sultan also had the 'customary presents' for Queen Catherine placed in his possession for him to take to her. Here one sees, as had happened in with the embassy of Pero Tafur in 1436 discussed above, how presents were used successfully as bribes to win the persons close to the sultan around, with the sultan, once mollified, granting his own presents to the queen, the envoy Thomas Ficard, and to one of his translators (Mas Latrie 1882, pp. 518–19; Hill 1940–1952, vol. 3, p. 735; Coureas 2016, pp. 369–70).

On his return to Cyprus in mid-December 1483 Ficard appeared before Quen Catherine at Larnaca, but without presenting the sultan's presents for her because these, loaded on some carts, were arriving in his wake. The fact that these unspecified presents, too bulky for him to bring himself, had been loaded on carts indicates that they were of great quantity and value. Ficard requested an adjournment until the following Wednesday, on which day the letters the sultan had sent to the queen as well as those of certain other eminent persons, were read out, while the sultan's presents were presented to her. Ficard had a copy of the sultan's letter translated from Arabic into Latin and dispatched to George Contarini so that the latter would be apprised of its contents. Ficard underlined the success of his mission by pointing out that on the day that he and his translator had received robes of honor the Neapolitan ambassador who had requested an audience with Sultan Qaitbay to obstruct Venetian business had been refused an audience. The presents sent to the sultan's close advisers had undoubtedly helped Ficard succeed in his mission (Mas Latrie 1882, pp. 519–21; Hill 1940–1952, vol. 3, p. 735; Coureas 2016, pp. 370–71). Yet at this time, King Ferrante of Naples had also sent the sultan an extremely impressive gift, a ship loaded with weapons, including sets of armor, helmets, brassards, lances, halberds, axes, swords, maces, culverins, other firearms, artillery pieces and gunpowder. Sultan Qaitbay was delighted with the present. Nevertheless, he must also have realized that Venice, with a much greater volume of trade in his domains than the kingdom of Naples and a powerful fleet, which Naples lacked, was a far more useful ally. The incident shows that gifts were a secondary factor in the success of a diplomatic mission, not the decisive factor (Behrens-Abouseif 2016, p. 106; Coureas 2016, p. 371).

The abdication of Queen Catherine in 1489 necessitated the dispatch of a Venetian embassy to Sultan Qaitbay to justify the resultant Venetian annexation of Cyprus. The first ambassador sent, Marco Malipiero, reached Cairo on 25 April 1489, bringing with him two years' tribute and presents in the form of fabrics of silk and camlet. It availed him little, for the sultan, angered by the Venetian seizure of Rizzo de Marino, a sworn enemy of Venice since the death of King James II but also a Mamluk ambassador, took the presents but refused to see him. He referred him to his *dawadar*, whom Marco declined to see given that he had been sent to have an audience with the sultan himself. Several months later, Venice decided to send another envoy to the sultan, Piero Diedo. Doge Agostino Barbadico in a letter of 10 September 1489 instructed him on reaching Egypt to first make the usual visit to the emir of Alexandria to present letters and to give him the presents customarily given. On securing an audience with the sultan, Piero Diedo would express hopes for peace regarding the Ottoman-Mamluk war in progress, the continuation of which could only harm Venetian interests. If, however, peace had not yet been concluded, Piero Diedo would simply wish the sultan prosperity, presenting the customary presents to him. After visiting the sultana and various high-ranking court officers, Piero Diedo would make representations to the sultan over the welfare of the Venetian merchants in his lands. He would raise the issue of Cyprus, presenting various arguments justifying its direct annexation, only if Marco Malipiero had failed to convince the sultan that the direct annexation of Cyprus by Venice was also in his own best interests. Towards the end of the doge's letter, Piero Diedo was instructed, were he to encounter Taghribirdi, the sultan's extremely influential interpreter who was a man of great ability, to grant him gifts secretly to facilitate the success of his mission (Mas Latrie 1852–1861, vol. 3, pp. 472–78; Hill 1940–1952, vol. 3, pp. 821–23; Rossi 1988, pp. 259–64; Coureas 2016, pp. 373–74).

On reaching Cairo on 7 December 1489 Piero Diedo obtained an audience with Sultan Qaitbay. Following negotiations in which Piero Diedo's secretary Giovanni Borghi and Taghribirdi took part, an agreement was reached in February 1490, whereby the sultan recognized direct Venetian rule over Cyprus in return for continued and regular payments of the tribute from the island. Piero Diedo, however, died in late February, so the negotiations were completed by Giovanni Borghi. On 9 March the sultan's representatives in Egypt accorded recognition of Venice's direct rule over Cyprus so long as the tribute continued to be paid regularly in the customary manner. On 27 March 1490, Giovanni Borghi received

the formal letter of the sultan's acquiescence to Venetian rule. It recounted the history of the negotiations, the death of Piero Diedo, the request that Cyprus be ruled directly from Venice, that the inhabitants be well treated, protected from their enemies, and should have their differences resolved, and that the doge's representative on Cyprus should have the tribute paid and sent to Egypt promptly and regularly. The sultan bestowed robes of honor to Giovanni Borghi, and via him robes to the doge, for the latter to bestow at his pleasure. Another robe of honor had been granted to Marco Malipiero, the ambassador sent from Cyprus, who was now sent back there with letters for the Venetian proveditor of Cyprus (Mas Latrie 1852–1861, vol. 3, pp. 478–83; Hill 1940–1952, vol. 3, pp. 823–24; Rossi 1988, pp. 269–71; Coureas 2016, pp. 374–75).

Finally, the sultan sent the doge via Giovanni Borghi a 'small present,' consisting of the following; one ampule of balsam, two horns of civet perfume, 25 boxes of theriac, 35 rotoli of aloe wood, 35 rotoli of benzoic resin, nine bolts of silk, five pieces of cinnabar, 25 plates of porcelain, eight dishes of porcelain, 100 pieces of sugar and two boxes of powdered white sugar. The composition of this gift reflects the fact that the gift exchanges of the Mamluks with Venice were among the most significant in Mamluk diplomacy with European powers. Recorded Mamluk gifts to Venice, dating mainly to the fifteenth century, consisted generally of spices, scents, textiles and porcelain, much as the gift described above. Some relevant instances are Sultan Jaqmaq's gift in 1442 to the Venetian doge Foscari, a gift of either Sultan Al-Mu'ayyad Ahmed or Sultan Khushqadam in 1461 to the Venetian doge and the gift Sultan Qaitbay sent in 1473 to Doge Nicolo Tron (Mas Latrie 1852–1861, vol. 3, p. 483; Rossi 1988, p. 270; Behrens-Abouseif 2016, pp. 107–8).

By the end of the fifteenth century, however, the composition of the Mamluk gifts sent to the Venetians had changed somewhat. In 1499 Sultan Al-Zahir Qansuh included a silver gilded saddle, a saddle of crimson velvet with silver accoutrements and horse textiles, a blanket for the horse's body and a silk gold-embroidered caparison. Similarly, the gifts that the incumbent Sultan Qansuh al-Ghawri, the penultimate Mamluk sultan, sent in 1503 to the Venetian governor of Cyprus included a saddle of gilded silver, a horse-blanket made of gold and velvet along with more standard items such as a silk gown lined with ermine and other textiles, ten vessels of Chinese porcelain, one horn of civet perfume, aloe wood and benzoic resin, and ten boxes with containers of theriac (Behrens-Abouseif 2016, p. 108). The gift was sent shortly after the Portuguese in the Indian Ocean had attacked seven commercial ships transporting Mamluk merchants and their goods, forcing the Mamluks to alter their trade itineraries and buy spices from Ceylon, Sumatra and Malakka instead. The Venetians were also alarmed by the Portuguese penetration of the Red Sea and the Indian Ocean and their resultant interference in the spice trade. This development compelled them to be concerned about their hitherto unchallenged predominance in Europe's international trade. The gifts the Mamluk sultan sent to the Venetian governor on Cyprus reflect his desire to ally with the Venetians against the Portuguese. The addition of silver saddles among the Mamluk gifts sent to Venice or to Venetian Cyprus implies an upgrade in Venetian status in Mamluk diplomacy (Behrens-Abouseif 2016, pp. 109 and 112).

## 5. Conclusions

From the above, it is clear that the Lusignan kingdom of Cyprus exchanged gifts regularly with its Muslim neighbors, the Mamluk sultanate of Egypt and the Turkish emirates of Southern Anatolia. The extant records are fuller regarding the exchange of gifts with the Mamluks. The dispatch and receipt of gifts normally took place within the context of diplomatic missions, with the Cypriot envoys or those of the Mamluks and the Anatolian Turks giving or receiving the presents. Such presents were given during negotiations to end a war, to celebrate the conclusion of a peace, to obtain the other side's goodwill, to placate its anger, or, especially in the case of the Mamluks, as tokens of vassalage. The presents given to the Lusignan rulers and their envoys by the Mamluk sultans or the Anatolian emirs are in general better documented than those that the Lusignans gave to the sultans or emirs. This shows that the Christian sources from which most information on the exchange of

gifts originates were more interested in the gifts given to the Lusignan rulers than in those that these rulers sent to the Mamluks or the Turkish emirs. In general, gifts were successful only when the parties exchanging them were already predisposed to conclude a peace, support a candidate for the throne of Cyprus or avert hostilities. Otherwise, they might be accepted for their value or as diplomatic tokens but failed in their intended purpose. The bestowal of gifts could influence the successful outcome of a mission but it did not determine it, gifts could be accepted even when a diplomatic mission failed.

**Funding:** This reseache received no external funding.

**Institutional Review Board Statement:** Not applicable.

**Informed Consent Statement:** Not applicable.

**Data Availability Statement:** All the research date comes from published sources and can be found in the bibliography.

**Conflicts of Interest:** The author declares no conflict of interest.

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
