# Peer review of "The Exchange of Gifts between Christians and Muslims on Lusignan and Venetian Cyprus 1192–1517"

_religions, doi:10.3390/rel14091163_

Round 1
Reviewer 1 Report
The discussion is persuasive and brilliant. I agree with the ending statement "The bestowal of gifts could influence the successful outcome of a mission but it did not determine it, gifts could be accepted even when a diplomatic mission failed".
However, It lacks a discussion about the power of gift in the ancient societies, a well-known topic after the classical work of Marcel Mauss, "The Gift: The Form and Reason for Exchange in Archaic Societies", London 2002 (new edition) [= Essai sur le don, Paris 1950]. In my opinion, a preliminary discussion would be helpful to improve the paper.
Author Response
Please note I have taken your comments into account. I have added a section to the introduction on page 2 of my paper, with reference to the monograph of Marcel Mauss and a bibliographical reference on page 14. These additions have been highlighted in green, please examine my revised paper.

Reviewer 2 Report
I urge the author to strengthen the analysis. As of now the article reads like a list of the gifts exchanged among medieval Eastern Mediterranean powers to broker peace, make allies, etc. This list is useful, but the conclusion is weak, basically this: sometimes the gifts worked, sometimes they did not. But there is no further analysis. Why did some work and not others?
The article could well do with a professional copyeditor. There a good many typos, a few infelicities of style and a few difficult (incomplete, incomprehensible) sentences.
Author Response
Please not that I have taken your comment into account. I have added sentences throughout this paper to strengthen the analysis. In addition, I have corrected the typos and stylistic infelicities. Furthermore, I have broken up sentences that were too long and therefore hard to understand. Please find these changes in the revised paper, they havge all been highlighted in yellow.

Round 2
Reviewer 2 Report
Oe more proofread would be of some value. But the submission has been improved considerably.